# Immunoexpression of Relaxin and Its Receptors in Stifle Joints of Dogs with Cranial Cruciate Ligament Disease

**DOI:** 10.3390/ani12070819

**Published:** 2022-03-23

**Authors:** Brunella Restucci, Mariafrancesca Sgadari, Gerardo Fatone, Giovanni Della Valle, Federica Aragosa, Chiara Caterino, Gianmarco Ferrara, Gert W. Niebauer

**Affiliations:** Department of Veterinary Medicine and Animal Production, University of Naples “Federico II”, 80137 Naples, Italy; restucci@unina.it (B.R.); mariafrancesca.sgadari@unina.it (M.S.); giovanni.dellavalle@unina.it (G.D.V.); federica.aragosa@unina.it (F.A.); chiara.caterino@unina.it (C.C.); gianmarco.ferrara@unina.it (G.F.)

**Keywords:** canine, cranial cruciate ligament rupture, relaxin, relaxin receptors

## Abstract

**Simple Summary:**

Spontaneous cranial cruciate ligament rupture is one of the most frequently encountered joint diseases in dogs, often leading to disabling chronic progressive osteoarthritis. The cause of the progressive intra-articular collagen matrix degradation, leading to tear and mechanical failure, is unknown. A variety of contributing factors has been found, however, an initiating mediator triggering the collagen degrading cascade remains to be identified. Our finding of strong relaxin- and relaxin receptor expression on intra-articular target tissues, such as on ligament fibrocytes and synovial membranes, renders relaxin a candidate for pathogenetic involvement, for collagen lysis, and progressive ligament fiber disruption. If confirmed, this opens the way for medical treatment of the disease in its early stages. In addition, further proof of relaxin involvement in canine osteoarthritis and ligament rupture would constitute a useful spontaneous animal model for human disease.

**Abstract:**

The etiology of spontaneous cranial cruciate ligament rupture in dogs is unknown despite being one of the most impacting orthopedic diseases in dogs. Numerous studies have contributed to the understanding of a multifactorial pathogenesis, this, however, without identifying a pivotal link to explain progressive collagen degeneration and osteoarthritic changes. In human medicine, recent reports have identified relaxin as a triggering factor in ligament ruptures in knee and metacarpal joints. We thus hypothesized that relaxin might also play a role in canine cruciate ligament rupture. Relaxin’s primarily known property is connective tissue remodeling through collagenolysis. We therefore investigated relaxin and its cognate receptors LGR7/LGR8 in 18 dogs with cranial cruciate ligament disease (CCLD) and compared them to a group of dogs with normal stifle joints. Applying immunohistochemistry (IHC), double immunofluorescence (dIF), and western blot analysis (WB), we found strong and significantly increased expression of both relaxin and its receptors in ruptured cruciate ligaments, and in synovial membranes. Pattern of immuno-staining on dIF strongly suggests relaxin binding to primed receptors and activation of signaling properties, which in turn may have affected collagen matrix metabolism. Thus, in canine cranial cruciate ligament disease, relaxin/receptor signaling may be a primary trigger for collagen fiber degradation and collagen lysis, eventually followed by ligament rupture.

## 1. Introduction

In cranial cruciate ligament disease (CCLD) a “non-contact” injury leads to rupture of a previously altered cranial cruciate ligament (CCL). The etiology of this spontaneous structural failure is still unknown, despite being one of the most impacting lesions in veterinary orthopedics. The nature of the process, which weakens the CCL progressively through collagen matrix degradation and eventually causes failure, still needs to be identified [1,2]. Multifactorial events including cellular [3], humoral [4], and metabolic [5] pathogenetic mechanisms have been implicated to contribute to the collagenous tissue degeneration [6,7]: upregulation and persistence of inflammatory mediators [8,9,10], especially matrix metalloproteinases (MMP) [11,12,13,14], immune and autoimmune reactivity [15,16,17,18,19,20] hypo-vascularity [21,22], biomechanic—[23,24,25,26], and genetic factors [27,28,29] have been implemented, among others. Matrix metalloproteinase, especially collagenase activation, seems to be a key factor involved in ligament degeneration and rupture [6,11,12,13,30]. However, signaling pathways and the chain of events leading to MMP activation within the cruciate ligament are ill understood [30,31]. Relaxin, a member of the insulin-like peptide family, is a potent MMP up-regulator, effectuating collagen lysis by binding to its cognate cellular receptors LGR7 and LGR8 [32]. Originally known as a facilitator of parturition in mammalian species, relaxin is now recognized as an effector in the signaling pathways of different organ systems in both males and females [33]. In connective tissues, relaxin mediates and modulates fibrolysis and collagenolysis [34]. This is a physiologic and reversible process in target tissues during parturition [35,36]. However, there is increasing evidence of relaxin involvement in processes of progressive connective tissue degradation: in humans, relaxin has been shown to cause joint laxity and ligament tears such as in the carpometacarpal joint [37,38,39], the knee [40], and the hip joint in women [41]. Cyclic relaxin surges in female athletes, enhancing knee joint laxity and exposing this group to cruciate ligament rupture [40]. In a guinea pig model, cruciate ligaments lost tensile strength when challenged with relaxin [42]. Although relaxin-related connective tissue laxity, whether physiologic or pathologic, is more common in females [42,43,44], relaxin is also expressed as the signaling peptide in males of many species [45,46]; for instance, admixed to sperm, relaxin mediates ovoid penetration [47]. There is evidence that in canines, relaxin is expressed in the prostate and excess levels and/or receptor activation are likely involved in connective tissue weakening processes such as in hernia formation [48,49,50]. That hormone-related mechanisms may be linked to CCLD has been previously suggested, however, without considering relaxin involvement [51]. Thus, and based on the cited findings of relaxin-related ligament alteration in man and hernia formation in dogs, we hypothesized that relaxin may be a factor in canine CCL degeneration. To that effect, we investigated immunoexpression and immunolocalization of relaxin and its cognate receptors LGR7 and LGR8 in CCL’s and in joint membranes of dogs with and without CCLD.

## 2. Materials and Methods

Samples for histological examination were obtained from 18 client owned dogs with CCLD, and from seven age- and body weight-matched fresh canine cadavers with normal stifle joints and intact CCL’s, euthanized for unrelated reasons. Owner consent for sampling was obtained for dogs in either group. All patients underwent routine surgical joint-stabilizing procedures including, and according to institutional protocol [52], medial mini-arthrotomy for the removal of torn ligament fragments, and when indicated, partial meniscectomy and excision of hypertrophic joint capsule. Tissue samples were divided into aliquots; for histology, specimens were fixed in 10% neutral buffered formalin, embedded in paraffin, and sections were stained with hematoxylin-eosin; for immunohistochemistry (IHC), double immunofluorescence (dIF), and western blot analysis (WB), aliquots were quick frozen in liquid nitrogen and stored at −80 °C. Frozen sections were processed using the streptavidin-biotin peroxidase method (LSAB Kit DAKO™, Glostrup, Denmark).

### 2.1. Immunohistochemistry

For IHC, sections were dried at room temperature for 1 h and fixed in acetone at 4 °C for 3 min. Endogenous peroxidase was blocked with hydrogen peroxide 0.3% and dehydrated in absolute methanol for 15 min at room temperature. Primary antibodies were diluted 1/100 in phosphate-buffered saline (PBS) and incubated overnight at 4 °C. For immunolabeling, negative control sections were incubated with normal serum IgG (Dako™) instead of the primary antibody. Samples of canine uterus in meta-estrus and of canine placenta were used as a positive control. A mixture of biotinylated anti-mouse and anti-rabbit immunoglobulins (LSAB kit; Dako™) in PBS was used as secondary antibodies and applied for 30 min. After washing in PBS, the sections were incubated for 30 min with streptavidin conjugated to horseradish peroxidase in Tris-Cl buffer containing 0.015% sodium azide (LSAB kit; Dako™). Diaminobenzidine tetrahydrochloride was used as the chromogen and hematoxylin as the counterstain for immunolabeling detection. Primary antibodies were directed against relaxin (rabbit polyclonal anti relaxin 2/RLN2 antibody Abcam™ AB232707), relaxin receptor 1 (rabbit polyclonal anti RXFP1/LGR7 Novus Biologicals™ NBP2-23674), and relaxin receptor 2 (rabbit polyclonal RXFP2/LGR8 Novus Biologicals™ NLS4751).

### 2.2. Double Immunofluorescence

For dIF, a two-color immunofluorescence staining method was applied in which primary antibodies were coupled as follows: relaxin with LGR7 and relaxin with LGR8. The pre-treatment steps and procedure were the same as for immunoperoxidase labeling. The rabbit anti-relaxin antibody was diluted 1:10 in the same diluent used for immunoperoxidase labeling and applied overnight at 4 °C. Slides were washed three times in PBS and incubated with tetramethyl rhodamine isothiocyanate (TRITC)-conjugated secondary goat anti-rabbit antibody, diluted 1:100 in PBS for two hours at room temperature. After rinsing three times in PBS, sections were incubated overnight at 4 °C with rabbit anti-RXFP1/LGR7 and rabbit anti-RXFP2/LGR8 antibodies, diluted 1:10, then rinsed three times in PBS and incubated for two hours at room temperature with fluorescein isothiocyanate (FITC)-conjugated secondary goat anti-rabbit antibody (Chemicon™, Germany) diluted 1:100 in PBS. Slides were rinsed with PBS and embedded in Fluorescent Mounting Medium (Dako™). A laser scanning microscope (Leica™ Microsystems, Germany) was used for scanning and photography. Rabbit polyclonal anti-relaxin antibody bound to TRITC was illuminated at 543 nm and read with a 560 nm band pass filter. Rabbit polyclonal RXPF1 and RXFP2 antibodies bound to FITC were illuminated at 488 nm and read with a 505–560 nm filter. Two-channel frame-by-frame multitracking was used for detection to avoid crosstalk signals. The different frames were scanned separately, and the optical path for excitation and emission of each scan was set up according to the manufacturer’s instructions.

### 2.3. Western Blot

Western blot analysis was performed on four ruptured CL samples and on five normal CL as controls; only well preserved and quantitatively sufficient samples were chosen, and gel-electrophoresis was repeated three times. Frozen samples were homogenized in ice-cold RIPA buffer (50 mM Tris-HCl, pH 7.5, 150 mM NaCl, 1% Triton X-100, 1 mM EDTA, deoxycholate 0.25%), admixed with phosphatase and protease inhibitor cocktails (Sigma-Aldrich, Milan, Italy) using TissueLyser machinery (Qiagen, Milan, Italy), according to the manufacturer’s protocol. The lysates were then centrifuged at 3500× *g* for 30 min at 4 °C. Total proteins were quantified by the Bradford assay (Bio-Rad™, Segrate, Milan, Italy). Equal amounts of protein lysates were separated by 12% SDS-polyacrylamide gel electrophoresis and then transferred to polyvinylidene difluoride (PVDF) membranes (Trans-Blot^®^ Turbo™ Mini PVDF Transfer Packs—Bio-Rad™) using Transblot Turbo apparatus (Bio-Rad™, Bio-Rad Laboratories, Segrate, Milan, Italy). The membranes were blocked with 5% non-fat dry milk in Tris buffered saline (10 mM Tris-HCl, pH 7.4, 165 mM NaCl) with 0.1% Tween (TTBS), for 1 h at room temperature and incubated overnight at 4 °C with the primary antibodies at the indicated dilutions: anti-relaxin2/RLN2 (Abcam, ab232707, rabbit polyclonal, 1:1000 dilution), anti-relaxin R1/LGR7 (Novus Biologicals™, NBP223674, rabbit polyclonal, 1:1000 dilution), anti-relaxin R2/LGR8 (Novus Biologicals™, NLS4751, rabbit polyclonal, 1:1000 dilution) and for glyceraldehyde 3-phosphate dehydrogenase (GAPDH, Santa Cruz™, sc-47724, mouse monoclonal, 1:1000 dilution) as normalization. After three washing steps of 10 min, appropriate peroxidase-conjugated secondary antibodies of anti-mouse IgG, HRP-linked (Cell Signaling Technology™, Antibody #7076, 1:2000 dilution), were applied for 1 h at room temperature at 1:2000 dilution. Following a further three washings in TTBS, bound antibodies were visualized by enhanced chemiluminescence with Clarity ™ Western ECL Blotting Substrate (Bio-Rad™, Segrate, Milan, Italy). The glyceraldehyde 3-phosphate dehydrogenase antibody was used as quality control to check even protein loading.

### 2.4. Histologic Scoring of Immunoreactivity

Immunostaining was graded by standard methods. On each IHC slide on 10 random visual fields at 400× magnification, by two independent pathologists (authors BR and MS), under a light microscope (Nikon E600; Nikon™, Tokyo, Japan), in a blinded semi-quantitative manner, staining was evaluated as follows:

Grading: four grades, corresponding to the percentage of immuno-stained cells: Grade 0: absence of stained cells; Grade 1: <10%; Grade 2: 10–30%; Grade 3: 31–60%; Grade 4: >60%; staining intensity was classified as weak (1), moderate (2), or strong (3).

Then, for each sample, a combined immunoreactivity score (IRS) on a scale between 0 and 12 was calculated by multiplying the values of these two classifications (grade of stained cells x staining intensity), as published [53].

### 2.5. Statistical Analysis 

All data were expressed as mean ± standard error of the mean (SEM). Comparisons were made using Student’s *t*-test or one-way analysis of variance followed by Tukey’s post hoc test, where appropriate. Analyses were performed using GraphPad Prism v7.0 software (La Jolla, CA, USA), and differences were considered statistically significant when *p* < 0.05. Power analysis using the online tool (https://clincalc.com/stats/samplesize.aspx, accessed 10 January 2022) was performed, confirming the appropriate minimum sample size for immunohistochemical results with an IRS CCL between 2.42 and 6.38 and an IRS SM between 2.8 and 7.5.

## 3. Results

A summary of the results is presented in Table 1. Of the seven dogs in the control group (CCLc, see Table 1a, there were three intact males and four neutered females; the medians for age were eight years (range 4–10) and for weight 20 kg (range 7–36). Of the 18 dogs with CCLD (Table 1b, CCLr), one had a partial ligament rupture and all others had completely ruptured cranial cruciate ligaments (CCLr). Five were intact males, five intact females, and eight were neutered females. Medians for age were six years (range 2–15) and for body weight 17.5 kg (range 5–44). In addition to tissue samples of cruciate ligaments, synovial membrane was sampled in five of the seven controls and in 14 of the 18 dogs with CCLD.

All dogs were treated surgically with different techniques; the owner-reported times between injury (spontaneous ligament rupture) and sampling was one month on average, indicating that patients had developed secondary osteoarthritis (OA) at the time of surgery. Additionally, marked inflammatory changes were evidenced by proliferative and often villous synovitis, with characteristic largely mononuclear infiltrates and synovioblast proliferations (see histologic description below). The number of cases with meniscal damage (eight out of 18), which was noted intraoperatively and listed in Table 1b, reflects the general pattern of joint derangement in CCLD.

### 3.1. Histologic Features of CCL and SM Samples

In all control ligaments (CCLc), collagen fibers were intact and longitudinally orientated, containing few fibrocytes that were row-like arranged (Figure 1a). In CCLr, there was a loss of longitudinal orientation, fibrillation, and proliferation of fibroblast-like cells in 14 of 18 cases. In 11 of them, there were also diffuse lymphocytic infiltrates present, admixed with a few polymorphonuclear granulocytes (PMNs) (Figure 1b). Synovial membrane samples in the control group (SMc) exhibited a normal monolayer of synoviocytes/histiocytes and fibrocytes on a loose connective tissue layer (Figure 1c). Synovial membranes of dogs with CCLr showed polystratification, sometimes forming villi due to synovioblast proliferation, diffuse or nodular infiltration of lymphocytes, plasma cells and PMNs as well as numerous, small vessels reflecting neovascularization (Figure 1d).

### 3.2. Immunohistochemistry

#### 3.2.1. Cranial Cruciate Ligaments (CCL)

In all seven normal ligaments (CCLc), ligamentous fibrocytes expressed relaxin weakly (7/7 100%, mean IRS = 2.62 ± 0.59 SE, range 1–4), with a largely diffuse cytoplasmatic expression pattern (Figure 2a). In the sample group (CCLr), relaxin immunostaining was strong throughout (18/18 100%, mean IRS = 5.55 ± 0.48 SE, range 4–9) and was characterized by large numbers of small cytoplasmic granules (Figure 2b). 

LGR7 receptors in normal ligaments (CCLc) were expressed by fibrocytes weakly in four cases and moderately in the remaining three; immunoreactivities were localized on the fibrocyte cell membranes (CCLc 7/7 100%, mean IRS = 2.42 ± 0.57 SE, range 1–4) (Figure 2c). In all damaged ligaments, receptor expression was 2.5-fold stronger on the average (CCLr 18/18 100%, mean IRS = 6.88 ± 0.51 SE, range 4–9); and immunoreactivity pattern was granular and intracellular (cytoplasmic) (Figure 2d).

LGR8 receptor expression was found to be similar to LGR7 expression in terms of strength and immunostaining pattern on and in fibrocytes; in CCLc 7/7 100%, mean IRS = 2.57 ± 0.42 SE, range 1–4, and in all CCLr (18/18 100%, mean IRS = 6.33 ± 0.49 SE, range 4–9). In CCLc, LGR8 immunoreactivity was characterized by fine granules diffused in the cytoplasm (Figure 2e) while in CCLr, LGR8 expression was more than 2-fold stronger, but cytoplasmatic diffuse (not granular) (Figure 2f). 

There was no significant difference in relaxin and/or receptor expression between dogs of different sexes; however, a trend was seen in the expression of relaxin that was stronger in ruptured ligaments of neutered females compared to males (mean IRS males 4.8; mean IRS neutered females 6.625 *p* = 0.1).

#### 3.2.2. Synovial Membranes (SM)

In all five controls (SMc) relaxin was only weakly expressed in synoviocytes in a membranous pattern, and in vascular endothelial cells (5/5 100%, mean IRS = 2.8 ± 0.58 SE (range 1–4) (Figure 3a), but strongly expressed in all 14 samples of dogs with CCLr (SMr) (14/14 100%, mean IRS = 6.57 ± 0.6 SE, range 2–9). The expression was more than 2-fold stronger than in the controls on average and diffused with a largely intracellular staining pattern (Figure 3b).

Weak LGR7 receptor expression was found in the five control membranes (SMc) (5/5 100%, mean IRS = 3.5 ± 0.5 SE, range 2–4), with staining restricted largely to synoviocytes and vascular endothelial cell membranes (Figure 3c). On the other hand, all 14 synovial membranes of dogs with CCLD (SMr) (14/14 100%, mean IRS = 7.5 ± 0.58 SE, range 4–9) expressed the receptor 2-fold stronger than the controls and largely with a membranous staining pattern (Figure 3d).

LGR8 receptor immunostaining exhibited a pattern and strength similar to the LGR7 expression in both, the controls and samples, respectively; SMc (5/5 100%, mean IRS = 3.6 ± 0.74 SE, range 2–6) (Figure 3e) and SMr (14/14 100%, mean IRS = 6.78 ± 0.42 SE, range 4–9) (Figure 3f).

Immunoreactivity scores of IHC data in the controls and samples are summarized and plotted in Figure 4.

### 3.3. Double Immunofluorescence (IF) Staining

In control ligaments (CCLc), co-localization of relaxin and the LGR7 receptor was absent (Figure 5a); in contrast, a strong relaxin/LGR7 co-localization staining on fibrocyte cell membranes was observed in ligament sections of CCLr, indicating relaxin/receptor binding (Figure 5b). 

In contrast, a cytoplasmatic relaxin/LGR8 co-localization was present in the controls (CCLc) (Figure 5c), while in CCLr, only a few cells expressed a membranous relaxin/LGR8 co-localization (Figure 5d).

In control synovial membranes (SMc), no co-localization of relaxin and LGR7 was observed; both targets were stained separately (Figure 5e). In contrast, in CCLr, strong relaxin expression as well as co-localization of relaxin/LGR7 was found, indicating the binding of relaxin to its cognate receptor in altered synovial membranes (Figure 5f).

Relaxin/LGR8 binding on the control membranes (CCLc) was evidenced by a moderate, largely cytoplasmatic co-localization (Figure 5g), while a strong relaxin/LGR8 co-localization staining was found in SMr, similar to the expression pattern of relaxin/LGR7 (Figure 5h).

### 3.4. Western Blot Analysis

In remnants of ruptured ligaments and in the controls, immunoreactive relaxin, LGR7, and LGR8-bands with the expected molecular weight were present, confirming cross-reactivity of the used antibodies in the canine species (Figure 6). An increased expression of relaxin, LGR-7, and LGR-8 in CCLD was therefore also demonstrated by WB. For two out of the three proteins, WB confirmed the expression values obtained in IHC and dIF. Immunoreactive bands for relaxin, LGR7, and LGR8 resulted in all samples with different signal intensity; in particular, densitometry confirmed the previously found significantly higher expression of relaxin and LGR7 in the CCLr samples (Figure 6a,b). However, the LGR8-bands expressed a higher protein density in the five ligament control samples compared to the four ruptured ligament samples, which is incongruent with the expression results obtained by immunostaining in IHC and dIF (Figure 6c).

## 4. Discussion

In the studied cohort of 18 randomly chosen dogs with CCLD, expression of relaxin and its cognate receptors LGR7 and LGR8 was significantly increased in damaged ligaments and in reactive joint membranes, compared to the corresponding normal structures of seven age and weight-matched control dogs. This finding is novel in the CCLD of dogs. Beyond this, nearly no data are available on relaxin in canine joints, whether normal or altered. Only one study suggested that in puppies, relaxin absorption through maternal milk may trigger hip joint laxity and later hip joint dysplasia [54].

In humans, on the other hand, relaxin has been implemented in joint disorders such as in tearing of the oblique volar ligament in the trapeziometacarpal joint [38] and in anterior cruciate ligament tears in female athletes [43,55]. In our cohort of dogs with CCLD, there were also more female (13) than male dogs (five), which is in accordance with known epidemiologic data in dogs: there is a 2-fold increased risk for developing cruciate ligament rupture in female dogs [56]. In the joint tissues we investigated, relaxin and relaxin receptors were nearly equally expressed in males and females. However, there was an indication of stronger relaxin expression in neutered females, compared to male dogs; this observation, which lacks statistical significance, merits further investigation. That relaxin also has paracrine signaling functions in males is now a generally accepted concept [45,57]; this has also been shown in the canine species: relaxin expression in prostate glands and connective tissues has previously been documented in pathologies typical for male dogs such as in perineal hernia [48,49]. Thus, the finding of relaxin expression in stifles of dogs of both sexes including desexed females finds explanation based on the growing knowledge of relaxin’s sexual dimorphism.

Although LGR7 and LGR8 receptors in the cruciate ligaments of dogs have not been studied previously, the presence and distribution of these receptors in normal target tissues (ligaments and tendons) have been described in humans and in laboratory animals [37,58]. In our dogs with CCLD, there was a significant increase in the expression of both cellular receptor types on cruciate ligaments and joint membranes. As shown by dIF, the co-localization of relaxin and its cognate receptors indicates relaxin–ligand binding to the activated receptors LGR7 and LGR8. The binding pattern of the hormone/receptor pairs on ligamentous fibrocytes and fibrocyte-like cells was membranous and cytoplasmatic, likely indicating transmembrane signal processing. The primarily known signals of the relaxin/receptor pair are collagenolytic in nature, largely via activation of MMP pathways [34,59] and downregulation of tissue inhibitors of MMPs [60], thereby inducing collagen matrix breakdown [32]; this, in turn, leads to collagen fiber crimp and fibrillation, likely even before mechanical failure manifests clinically [61]. This, because rupture occurs randomly in the degeneratively weakened ligament, and it may therefore well be that relaxin is already involved in collagen matrix degradation before complete failure has occurred and before secondary OA has developed. Since tissue sampling in those earlier stages of the joint disease can rarely be undertaken surgically because of the potential damage, future studies aided by arthroscopic sampling might help answer this question.

The mediators triggering synthesis of the LGR7/LGR8 receptors on cruciate ligament fibrocytes, and the mechanisms of receptor priming remain presently unknown in dogs; however, it has been shown in rats that sex-steroid hormones upregulate these receptors in ligaments [62]. That humoral factors are likely involved in CCLD has reached wide consensus [4,63]; thus, it may be assumed that the hormone relaxin reaches target tissues via humoral pathways, which, in the case of intra-articular tissues, depends on vascularity, vascular permeability, and synovial fluid diffusion and transfer. The CCL is a poorly vascularized intra-articular structure, especially in its central section [64,65], making it likely that relaxin’s bioavailability is mediated primarily by the synovial membrane and synovial fluid. Although we did not assay relaxin in the plasma or synovial fluids, the cellular and vascular inflammatory changes in the synovial membranes would facilitate relaxin transit via synovial fluid: all sampled synovial membranes exhibited increased micro-vascularity and characteristic, largely mononuclear inflammatory changes, in line with previous histologic findings that suggest a humoral component in the pathogenesis of CCLD [3,4]. In fact, we found strong endothelial expression of relaxin and its cognate receptors on synovial membranes and its capillary endothelia. Although neovascularization is a common feature of inflammation, relaxin itself has vascular growth factor-stimulating properties [66,67], and may therefore contribute to neo-angiogenesis in CCLD-related OA. Interestingly, local up-regulation of insulin-like growth factors in synovial fluids of stifle joints in dogs with CCLD and OA has been previously documented [68], and relaxin is structurally closely related to insulin-like growth factors, belonging to the same family of insulin-like peptides [57]. Functional comparisons based on structural similarities of the two peptide hormones must remain speculative in OA; however, the interaction of growth hormones and relaxin in connective tissue remodeling is well known [69] and it may well be that in CCLR, growth hormone and relaxin expression reflect both repair attempts and ligament degradation through collagen lysis. In fact, failure to heal and progressive ligament lysis after rupture is a characteristic feature of CCLR in dog: in non-treated cases or in those with surgery delayed by two months or more after ligament rupture, only stumps or small ligament remnants usually remain [24]. Our cohort included five such dogs from which samples of remnants were obtained between 60 and 70 days after injury, and all exhibited strong relaxin and receptor expression; this suggests that relaxin not only acts in early (partial rupture) but also in later phases of the collagenolytic process. The intriguing question, whether relaxin stands at the beginning of the collagenolytic cascade [34,59], that is, perhaps initially altering the structural integrity of the ligament, remains unanswered and awaits further studies.

Our study has limitations: the number of cases was relatively low, and the control cases, although age- and weight-matched, were cadaver specimens, and early stages or onset of CCLD might have remained undetected. This could also be an explanation, at least in part, for another possible flaw, which was the incongruency of LGR8 protein expression obtained by WB densitometry compared to the tissue/cellular expression of this receptor in IHC samples. That is, in WB, LGR8 protein bands were stronger in the five controls than in the four samples, whereas in IHC, LGR8 was, on average, significantly stronger expressed in the 18 samples compared to the seven controls, despite the samples and controls being from the same cohorts.

## 5. Conclusions

Relaxin and cognate receptor LGR7/LGR8 expression in stifle joints of dogs with spontaneous cranial cruciate ligament rupture is a new finding, indicating that relaxin-related collagen matrix degradation is a factor in CCLD. Relaxin likely reaches intra-articular target tissues via synovial fluid diffusion from neo-vascularized, osteoarthritic synovial membranes; relaxin/receptor signaling may contribute to ligamentous collagen degradation, even in early stages of the joint disease. If confirmed in further studies, targeted pharmacologic treatments may be developed.

## Figures and Tables

**Figure 1 animals-12-00819-f001:**
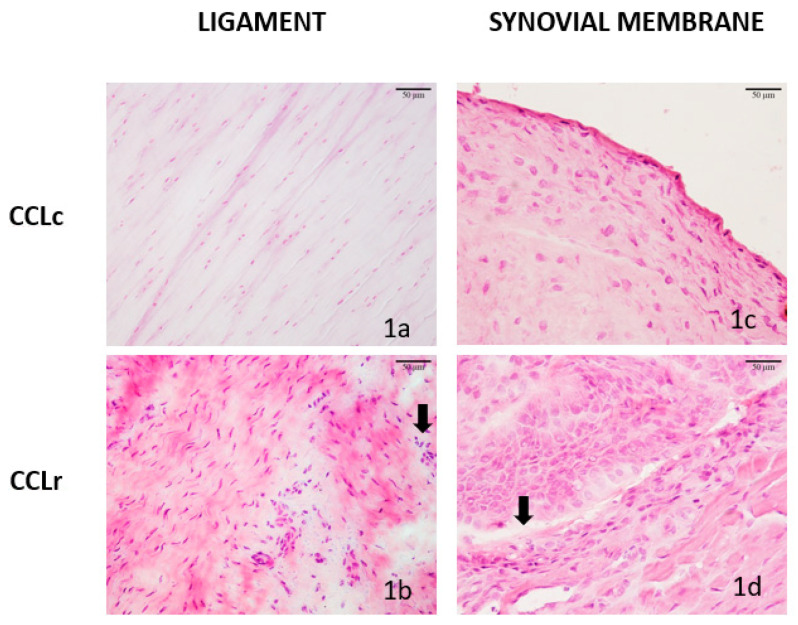
Histological features of canine cruciate ligaments (CCL) and synovial membranes (SM) in the controls (CCLc) and in remnants of ruptured ligaments (CCLr): (**a**) CCLc, case 3, collagen fibers longitudinally oriented with spindle-like fibrocytes in alignment with fibers; (**b**) CCLr, case 2, disorganization of collagen fibers, infiltration with lymphocytes and some polymorphonuclear granulocytes (arrow); (**c**) SM control (SMc) case 3, a monolayer of histiocytes and fibrocytes overlaying loose capsular tissue; (**d**) Altered SM (SMr) of case 2 with CCLr: thickening of synovial membrane, synovioblast proliferation, diffuse infiltration of lymphocytes and polymorphonuclear granulocytes (arrow), and angiogenesis, characterized by numerous small vessels. Hematoxylin and eosin staining, scale bars 50 μm.

**Figure 2 animals-12-00819-f002:**
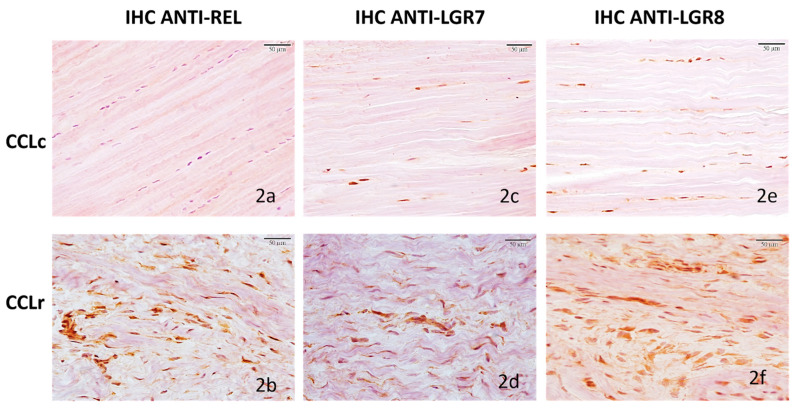
Immunohistochemical labeling of relaxin, LGR7, and LGR8 receptors in canine cruciate ligaments; controls (CCLc) and ruptured ligaments (CCLr): (**a**) CCLc, case 5, relaxin IHC, IRS = 1, weak, diffuse cytoplasmatic immunoreactivity in 9.5% of fibrocytes. (**b**) CCLr, case 17, relaxin IHC, IRS = 9, strong immunoreactivity, characterized by few, small cytoplasmic granules in 38.4% of fibroblast-like cells. (**c**) CCLc, dog 6, LGR7 IHC, IRS = 4, moderate immunoreactivity localized on cell membranes in 12.7% of fibrocytes. (**d**) CCLr, case 17, LGR7 IHC, IRS = 6, moderate immunoreactivity, characterized cytoplasmic granules in 38.5% of fibroblast-like cells. (**e**) CCLc, dog 6, LGR8 IHC, IRS = 2, moderate diffuse cytoplasmatic immunoreactivity in 10.7% of fibrocytes. (**f**) CCLr, case 17, LGR8 IHC, IRS = 9, strong immunoreactivity, diffuse and cytoplasmatic in 48.4% of fibroblast-like cells. Scale bars 50 μm.

**Figure 3 animals-12-00819-f003:**
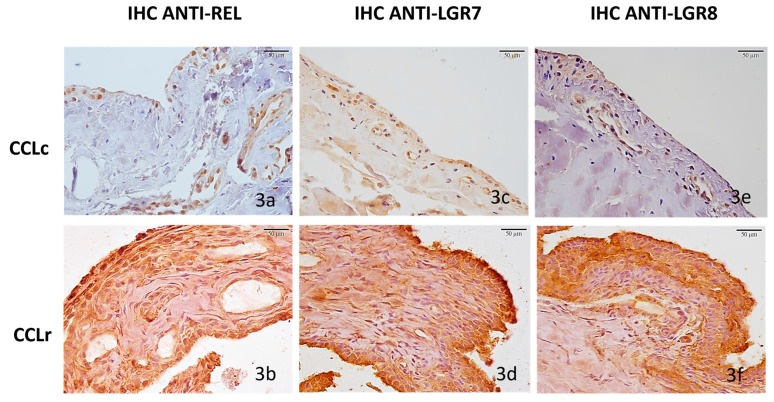
Immunohistochemical labeling of relaxin, LGR7, and LGR8 receptors in synovial membranes of control dogs (CCLc) and synovial membranes of dogs with CCLr. (**a**) Control case 3, relaxin IHC, IRS = 1, weak diffuse cytoplasmatic immunoreactivity in 8.7% of synoviocytes. Relaxin-immunostaining was also evident in vascular endothelial cells. (**b**) SMr of CCLr case 14, relaxin IHC, IRS = 9, strong diffuse cytoplasmatic immunoreactivity in 36.2% of synoviocytes. (**c**) Control case 3, LGR7 IHC, IRS = 2, moderate immunoreactivity on the cell membranes in 9.7% of synoviocytes, and on vascular endothelial cells (**d**) CCLr, case 14, LGR7 IHC, IRS = 9, strong membranous immunoreactivity in 33.9% of fibroblast-like cells. (**e**) Control case 3, LGR8 IHC, IRS = 2, moderate diffuse, cytoplasmatic immunoreactivity in 7.9% of synoviocytes. (**f**) CCLr, case 14, LGR8 IHC, IRS = 6, strong membranous immunostaining as well as diffused cytoplasmatic staining in 28.2% of fibroblast-like cells. Scale bars 50 μm.

**Figure 4 animals-12-00819-f004:**
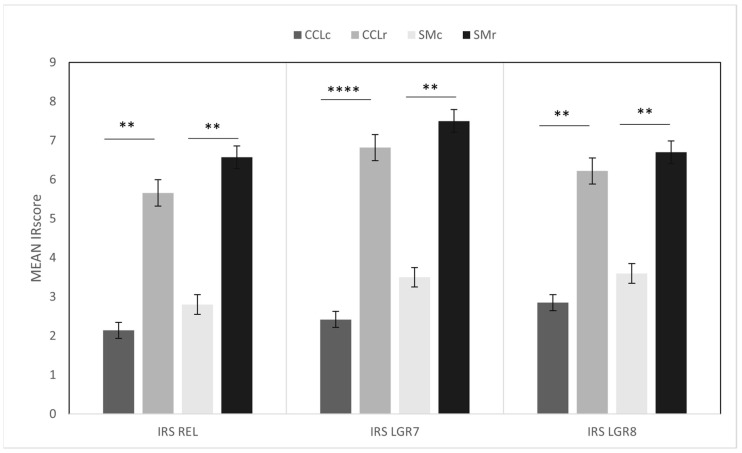
Bar graphs showing expression of relaxin (REL), LGR7, and LGR8 in synovial membranes and cruciate ligaments in the control group (*n* = 7), SMc, and CCLc columns, respectively, and in dogs with CCLD (*n* = 18), SMr and CCLr columns, respectively; values are expressed as mean immunoreactivity score (IRS) with standard error bars; significance levels of REL CCLc/CCLr *p* = 0.001 (**), LGR7 CCLc/CCLr *p* < 0.001 (****), LGR8 CCLc/CCLr *p* = 0.001 (**), REL SMc/SMr *p* = 0.02 (**), LGR7 SMc/SMr *p* = 0.001 (**), LGR8 SMc/SMr *p* = 0.003 (**).

**Figure 5 animals-12-00819-f005:**
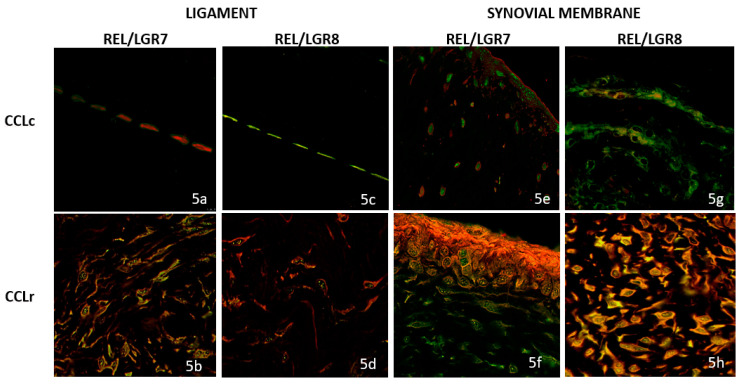
Double immunofluorescence (IF) staining: relaxin = cytoplasmic red IF; LGR7 = (membranous green IF) relaxin/LGR7 co-localization = membranous yellow IF, and relaxin/LGR8 = cytoplasmic yellow IF; for textual explanation see Results above; CCLc = control dog ligaments in (**a**,**c**), and normal synovial membranes in (**e**,**g**); CCLr ruptured ligaments in (**b**,**d**), and altered synovial membranes of dogs with CCLr in (**f**,**h**). Full original blots can be found in Appendix A.

**Figure 6 animals-12-00819-f006:**
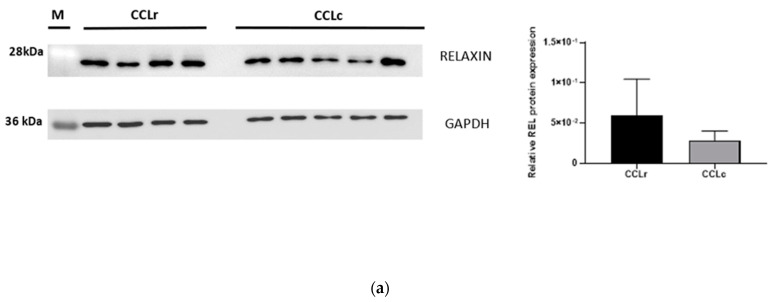
Western blot analysis of relaxin (**a**), LGR7 (**b**), and LGR8 (**c**) protein expression in four CCLr tissue samples and in five CCLc tissues. The bar graphics show the mean ± SEM of relative relaxin, LGR7, and LGR8 protein expression in the CCLr and CCLc samples. Plots show densitometric analysis of relaxin, LGR7, and LGR8 protein bands expressed as relaxin, LGR7, or LGR8/GAPDH densitometry ratio for each tissue sample. Note that GAPDH was identical solo for relaxin and LGR7; for LGR8, the same GAPDH was applied but on a different membrane. M = molecular weight marker.

**Table 1 animals-12-00819-t001:** The two columns to the right show the mean immunoreactivity scores (IRS) on IHC in cruciate ligaments and synovial membranes, respectively; in each column, the first vertical row of numbers represents relaxin-specific IRS, the second row LGR7-, and the third LGR8-specific IRS for each dog. Abbreviations: MP modified Maquet procedure; TTA tibial tuberosity advancement; TPLO tibial plateau levelling osteotomy; DE ANGELIS extracapsular suture stabilization; M male; F female; FS female sterilized, NA not available. (**a**) Seven control dogs without CCL damage (CCLc). (**b**) Eighteen dogs with CCLD (CCLr).

(**a**)								
**No. Case Control Group CCLc**	**Age (Years)**	**Breed**	**Body Weight (kg)**	**Sex**	**IRS IHC Ligaments REL; LGR7; LGR8** ***n* = 7**		**IRS IHC Synovial Membranes, REL *n* = 5; LGR7 *n* = 4; LGR8 *n* = 5**	
1	5	MIXED BREED	23	FS	1; 1; 1		NA	
2	10	MIXED BREED	7	M	2; 2; 2		3; 4; 6	
3	8	ROTTWEILER	36	M	2; 1; 4		1; 2; 2	
4	9	MIXED BREED	31	FS	4; 4; 4		4; NA; 2	
5	6	MIXED BREED	8	FS	1; 1; 2		4; 4; 4	
6	10	MIXED BREED	11	FS	2; 4; 2		NA	
7	4	ENGLISH BOULEDOGUE	20	M	3; 4; 3		2; 4; 4	
	Median 8 Range 4–10		Median 20 Range 7–36		$\bar{x}$ 2.62 SE 0.59 $\bar{x}$ 2.42 SE 0.57 $\bar{x}$ 2.57 SE 0.42		$\bar{x}$ 2.8 SE 0.58 $\bar{x}$ 3.5 SE 0.5 $\bar{x}$ 3.6 SE 0.74	
(**b**)								
**No. Case** **Sample Group CCLr**	**Age (Years)**	**Breed**	**Body Weight (kg)**	**Sex**	**Surgical Procedure**	**Time Since Injury (Days)**	$\bar{x}$ **IRS IHC Ligaments REL; LGR7; LGR8** ***n* = 18**	$\bar{x}$ **IRS IHC Synovial Membranes, REL; LGR7;LGR8** ***n* = 14**
1	8	MIXED BREED	25	FS	TPLO	UNKNOWN	4; 4; 4	6; 9; 6
2	4	LABRADOR RETRIVER	34	FS	MP TTA + partial meniscectomy	14	9; 9; 6	9; 9; 6
3	7	JACK RUSSEL TERRIER	6	M	DE ANGELIS	70	6; 8; 6	9; 4; 4
4	5	MIXED BREED	15	F	DE ANGELIS	21	9; 9; 9	9; 9; 9
5	5	CAIRN TERRIER	11	FS	DE ANGELIS	60	4; 6; 4	NA
6	11	YORKSHIRE	5	M	DE ANGELIS	60	4; 4; 6	NA
7	7	MIXED BREED	9	M	DE ANGELIS	30	4; 4; 4	4; 4; 6
8	5	MIXED BREED	18	FS	MP TTA+ partial meniscectomy	60	6; 9; 9	2; 6; 6
9	2	CORSO	44	F	MP TTA	30	4; 9; 9	4; 9; 9
10	15	JACK RUSSEL	6	FS	De ANGELIS	UNKNOWN	6; 6; 6	NA
11	7	BOXER	33	FS	DE ANGELIS	30	9; 9; 9	6; 9; 9
12	8	MIXED BREED	27	F	DE ANGELIS	15	4; 4; 4	6; 6; 6
13	8	MIXED BREED	41	F	MP TTA	30	4; 4; 4	9; 9; 9
14	7	BARBONE MEDIO	14	M	DE ANGELIS	10	4; 6; 6	9; 9; 6
15	2	CORSO	42	F	MP TTA	60	4; 9; 4	NA
16	5	AMERICAN STAFFORDSHIRE TERRIER	17	FS	MP TTA	30	6; 9; 6	6; 4; 6
17	5	AMERICAN STAFFORDSHIRE TERRIER	16	FS	MP TTA	30	9; 6; 9	4; 9; 6
18	5	AMERICAN STAFFORDSHIRE TERRIER	18	M	MP TTA	21	6; 9; 9	9; 9; 6
	Median 6 Range 2–15		Median 17.5 Range 5–44			Median 30 Range 10–70	$\bar{x}$ 5.55 SE 0.48 $\bar{x}$ 6.88 SE 0.51 $\bar{x}$ 6.33 SE 0.49	$\bar{x}$ 6.57 SE 0.65 $\bar{x}$ 7.5 SE 0.58 $\bar{x}$ 6.7 SE 0.42

## Data Availability

Not applicable.

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
