# Peer review of "Immunoexpression of Relaxin and Its Receptors in Stifle Joints of Dogs with Cranial Cruciate Ligament Disease"

_animals, 2022, doi:10.3390/ani12070819_

Round 1
Reviewer 1 Report
The manuscript entitles “ Immunoexpression of relaxin and its receptors in stifle joints of dogs with cranial cruciate ligament disease ” presents the relaxin/receptor signaling may be a primary trigger for collagen fiber degradation and collagen lysis, followed by ligament rupture eventually, in canine cranial cruciate ligament disease. Samples from clinical animals showed good research value. Based on these, I'm very interested in the results of this study. However, I still have some incomprehensible points which need to be answered by the authors.
- Why were negative control sections incubated with normal serum IgG 96 (Dako) instead of primary antibodies in Immunohistochemistry ?
- Why did not assay relaxin in plasma or in synovial fluids?
- It is well known that the interaction of growths hormone and relaxin in connective tissue remodeling. Why did not detect growth hormone?
- What is the reason for the incongruency of LGR8 protein expression obtained by WB densitometry?
5 The differences between different genders should be shown.
6 If the author has relevant research results on inflammation, it is suggested that the author provide supplement.
Author Response
please see attachement

Reviewer 2 Report
This paper thoroughly and convincingly evaluates the hypothesis that relaxin is involved in cranial cruciate ligament rupture in the dog.
Cranial or anterior cruciate ligament rupture is a common condition in pet dogs, and surgical repair is a common elective surgery in a busy companion animal hospital.
The introduction outlines the rationale and frames the work adequately, although it could be improved. The methods are particularly clear. The results map well to the methods, and the discussion is logically presented.
The conclusions are appropriate.
Thus reviewer particularly appreciated the systematic approach demonstrated here, and the including first and foremost of the presentation of the H&E histological description before presentation of the immunohistological and immunohistochemical work.
The tables and certain figures need much further work and current presentation reduces the quality of this paper. The presentation does not all match to the journal general instructions- see: https://academic.oup.com/jas/pages/General_Instructions
Comments to improve the paper
The introduction lays out the rationale for the hypothesis, however, this journal reaches a wider audience than those who are familiar with clinical conditions in dogs. This paper is of a level that is likely to be of interest to a non-veterinary audience, whose understanding of ligamentous damage is more related to human athletes. The authors should put in a sentence or two in the introduction to match the clear statement in the abstract to indicate the epidemiology/ presentation of anterior cruciate rupture in dogs for that audience.
There are some incorrect use of commas and language in the text. This reviewer notes that a grammar checker may have helped create some sentences- however, the grammar checker use does not always lead to readability.
Lines 54-55 (commas & readability)
Relaxin, is a member of the insulin-like peptide family. It is, Physiologically, it is a potent MMP up-regulator, effectuating collagen lysis by binding to its cognate cellular receptors LGR7 and LGR8.[32]
lines 63-65:
Suggest:
"and, in cruciate ligaments of female athletes, in whom cyclic humoral relaxin surges enhance knee joint laxity and thus expose to cruciate ligament rupture[40]"
is replaced with:
Cyclic relaxin surges in female athletes enhance knee joint laxity and expose this group to cruciate ligament rupture [40].
Line 65: Error- "Guinea Pig"
Line 71-73 - incorrect comma/missing article:
That hormone-related mechanisms, may be linked to CCLD has been suggested before, however, without suspecting the involvement of relaxin [50].
Line 192 "anamnestically". Although technically correct- this is not a commonly used word in English and would require a dictionary for most readers. "Owner-reported" may be better.
Line 337: Thumb?
There may be more, so this paper should be proofread further before publication.
Figures and Tables
Figures 1-3
Histological images should include a scale bar.
Tables 1a and 1b
These tables do not "stand-alone" in that they are not understandable without checking through the text of the paper. The table legends must provide much more clarity. As an example, it is not instantly clear what "IRS IHC LIGAMENTS REL; LGR7; LGR8" and the numbers associated with this actually mean.
The tables and legends should use a decimal point rather than a comma ie
xÌ„ 5,55 SE 0,48 should be 5.55±0.48
"Table 2" is a figure, not a table. These graphs need to be reproduced with better contrast and better-chosen differentiators. The CClr pattern is particularly difficult to look at. As black/greyscale is used in Figure 5, colour use should be consistent within the same paper.
Citation
The citation style does not appear to match the current journal style
Author Response
please see attachement

Round 2
Reviewer 1 Report
The author has answered some key questions.